# Novel origin of lamin-derived cytoplasmic intermediate filaments in tardigrades

Lars Hering[1,2*†], Jamal-Eddine Bouameur[3†], Julian Reichelt[1†], Thomas M Magin[3], Georg Mayer[1,2*]

[1]Department of Zoology, Institute of Biology, University of Kassel, Kassel, Germany; [2]Animal Evolution and Development, Institute of Biology, University of Leipzig, Leipzig, Germany; [3]Institute of Biology and Translational Center for Regenerative Medicine, University of Leipzig, Leipzig, Germany

**Abstract** Intermediate filament (IF) proteins, including nuclear lamins and cytoplasmic IF proteins, are essential cytoskeletal components of bilaterian cells. Despite their important role in protecting tissues against mechanical force, no cytoplasmic IF proteins have been convincingly identified in arthropods. Here we show that the ancestral cytoplasmic IF protein gene was lost in the entire panarthropod (onychophoran + tardigrade + arthropod) rather than arthropod lineage and that nuclear, lamin-derived proteins instead acquired new cytoplasmic roles at least three times independently in collembolans, copepods, and tardigrades. Transcriptomic and genomic data revealed three IF protein genes in the tardigrade *Hypsibius dujardini*, one of which (cytotardin) occurs exclusively in the cytoplasm of epidermal and foregut epithelia, where it forms belt-like filaments around each epithelial cell. These results suggest that a lamin derivative has been co-opted to enhance tissue stability in tardigrades, a function otherwise served by cytoplasmic IF proteins in all other bilaterians.

**\*For correspondence:** lars.hering@uni-kassel.de (LH); georg.mayer@uni-kassel.de (GM)

†These authors contributed equally to this work

**Competing interests:** The authors declare that no competing interests exist.

## Introduction

Tardigrades, also known as water bears, are microscopic invertebrates that live in marine, freshwater and semi-aquatic/limno-terrestrial environments (*Kinchin, 1994*; *Nelson, 2002*) (*Figure 1*). Although tardigrades have become renowned for their ability to survive extreme conditions (*Møbjerg et al., 2011*; *Ramløv and Westh, 2001*), including exposure to space (*Persson et al., 2011*; *Rebecchi et al., 2011*), only little is known about the actual mechanisms that allow their cells to resist severe mechanical stress caused by desiccation and freezing (*Møbjerg et al., 2011*; *Ramløv and Westh, 2001*; *Wright, 2001*; *Tanaka et al., 2015*; *Yamaguchi et al., 2012*). The integrity and plasticity of tardigrade tissues might be achieved by specialised cytoskeletal components, such as IF proteins, which are known to be essential for stress resilience of cells (*Herrmann et al., 2009*; *Coulombe and Wong, 2004*; *Kim and Coulombe, 2007*). While lamins, a group of IF proteins found in the nucleus, occur in most eukaryotes, including social amoebae (*Krüger et al., 2012*) and all metazoans (*Dittmer and Misteli, 2011*), cytoplasmic IF proteins are thought to have evolved from an ancestral *lamin* gene by duplication in the bilaterian lineage (*Erber et al., 1999*; *Herrmann and Strelkov, 2011*). Genomic and biochemical studies have revealed that the cytoplasmic IF proteins are present in all bilaterian taxa excluding arthropods (*Bartnik and Weber, 1989*; *Goldstein and Gunawardena, 2000*; *Erber et al., 1998*) [but see a contradictory report (*Mencarelli et al., 2011*) of a putative cytoplasmic IF protein in a collembolan]. The apparent loss of cytoplasmic IF proteins in the arthropod lineage might correlate with the acquisition of an exoskeleton (*Herrmann and Strelkov, 2011*; *Goldstein and Gunawardena, 2000*; *Erber et al., 1998*), which provides mechanical support to the arthropod skin. However, this hypothesis has never been tested,

**eLife digest** Different proteins exist to support the stability of animal cells. The intermediate filament proteins are an important example. One type – called lamins – stabilizes the nucleus (the structure within an animal cell that stores most of its DNA), while another forms scaffold-like structures in the rest of cell. The second type, referred to as "cytoplasmic" intermediate filaments, are not found in many hard-bodied creatures including insects and their closest relatives. This is probably because these animals, which are collectively known as arthropods, are instead supported by their tough external skeleton.

The soft-bodied animals called tardigrades (also known as water bears or moss piglets) are closely related to the arthropods. These microscopic animals can endure extreme environmental conditions such as freezing. The tardigrade's endurance is likely to require some way to stabilize the animal's cells. This might involve cytoplasmic intermediate filaments, but nothing was known about these proteins in tardigrades.

Now, Hering, Bouameur, Reichelt et al. have investigated if, and where, intermediate filaments are found in the cells of tardigrades. First, the complete set of active genes was analyzed for a species of tardigrade called *Hypsibius dujardini*; this revealed that three genes for intermediate filament proteins were active. Staining tissue slices or whole tardigrades with a marker that binds to intermediate filament proteins revealed that two of the three proteins were lamins and located within the nucleus. The third protein, which has been named "cytotardin", was found outside of the nucleus. However, unlike well-known cytoplasmic intermediate filaments, this protein did not form scaffold-like structures throughout the cell. Instead, cytotardin formed belt-like filaments that encircled each cell in the skin of the tardigrades.

Hering, Bouameur, Reichelt et al. then discovered that cytotardin seems to be more closely related to lamins than it is to cytoplasmic intermediate filaments. This suggests that cytotardin actually evolved from a tardigrade lamin and then acquired a new role in building filaments outside of the nucleus.

The fact that cytotardin is only found in the skin of the tardigrade and in those tissues that experience mechanical stress (for example, the mouth and legs) hints that it might help stabilize these cells. This could mean that the protein also helps these animals to resist extreme conditions. Further studies should focus on clarifying cytotardin's role in stabilizing cells, in particular if it is required for the tardigrades' tolerance to environmental stress.

as it is unknown whether or not onychophorans and tardigrades, the soft-bodied relatives of arthropods, possess cytoplasmic IF proteins.

## Results and discussion

To clarify whether or not onychophorans and tardigrades possess cytoplasmic IF proteins, we analysed Illumina-sequenced transcriptomes of five distantly related onychophoran species and the freshwater tardigrade *H. dujardini*. Although this model tardigrade (*Gabriel et al., 2007*; *Gross et al., 2015*) shows only a limited ability to tolerate desiccation (anhydrobiosis), it clearly survives immediate freezing (cryobiosis; see *Video 1*). Our analyses revealed only one putative IF protein transcript in each of the five onychophoran species but three in *H. dujardini*. Additional screening of the recently sequenced genome (*Boothby et al., 2015*; *Koutsovoulos et al., 2015*) of *H. dujardini* confirms that the identified transcripts correspond to the three potential IF protein-coding genes of this species. According to sequence comparisons with well-characterized IF proteins from humans and the nematode *Caenorhabditis elegans*, all corresponding proteins of the three identified genes share a similar α-helical rod domain organization with three coiled coil-forming segments (coil 1A, coil 1B, coil 2; *Figure 2A,B* and *Figure 2—figure supplements 1* and *2*) (review *Chernyatina et al., 2015*).This, in conjunction with the highly conserved intermediate filament consensus motifs (review *Herrmann and Aebi, 2004*) at the beginning and end of the rod domain of all three tardigrade proteins, classify them as intermediate filament proteins. The three tardigrade IF proteins further possess 42 residues in the coil 1B (*Figure 2B* and *Figure 2—figure supplement 1*)

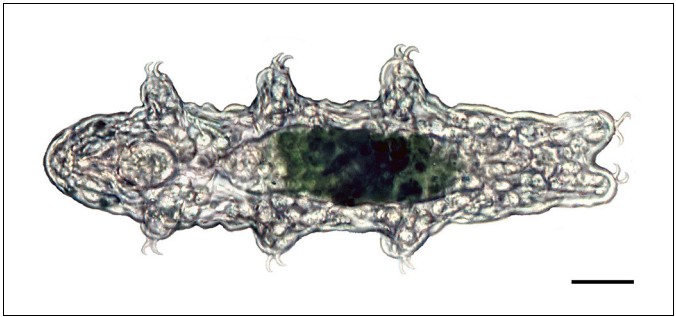

**Figure 1.** Light micrograph of a specimen of the tardigrade *Hypsibius dujardini* in dorsal view. Anterior is left. Scale bar: 20 µm.

— a feature that is shared between all eukaryote lamins and protostome cytoplasmic IF proteins but that must have been deleted from the ancestral cytoplasmic IF protein gene in chordates (*Herrmann et al., 2009*; *Peter and Stick, 2015*). In contrast to the similar organization of the rod domain, the flanking sequences vary between all three tardigrade IF proteins. One of them, which we named lamin-2, possesses all major regions known from other eukaryote lamins (*Herrmann et al., 2009*; *Dittmer and Misteli, 2011*; *Burke and Stewart, 2013*), including the nuclear localization signal (NLS) — which generally mediates the import of proteins into the nucleus (*Mical et al., 2004*) — the lamin-tail domain (LTD), and the carboxy-terminal CaaX motif (*Figure 2A, B*). In contrast, the second tardigrade IF protein (named lamin-1) lacks the CaaX box, whereas the third IF protein from *H. dujardini* is missing all three motifs, including the NLS, indicating this protein may instead localize in the cytoplasm rather than the nuclear lamina of *H. dujardini* cells. Since the structure of the latter resembles the domain composition of known bilaterian cytoplasmic IF proteins (*Figure 2—figure supplements 1* and *2*) we consequently named it cytotardin.

To firmly place the onychophoran and tardigrade IF homologs on the evolutionary tree, we reconstructed the phylogeny of broadly sampled metazoan lamin and cytoplasmic IF protein genes. In our phylogenetic analyses, all identified tardigrade and onychophoran sequences cluster within the bilaterian lamin clade, whereas none of them groups with cytoplasmic IF protein-coding genes (*Figure 3* and *Figure 3—figure supplements 1* and *2*). Surprisingly, the three identified IF sequences of *H. dujardini*, together with two sequences from *Milnesium tardigradum*, form a strongly supported monophyletic clade of tardigrade lamins (GTR+G: Bootstrap support BS=77, LG+G: BS=80) in our analyses (*Figure 3* and *Figure 3—figure supplements 1* and *2*). This implies that there were at least two duplication events in the tardigrade lineage that gave rise to *lamin-1, lamin-2* and *cytotardin* genes — consequently characterizing the tardigrade IFs (including cytotardin), for example, as co-orthologous to nematode lamins rather than orthologous to nematode cytoplasmic IFs (*Figure 3* and *Figure 3—figure supplements 1* and *2*). Our results further show that the isomin sequence of the collembolan *Isotomurus maculatus* does in fact cluster with other identified collembolan transcripts (GTR+G: BS=74, LG+G: BS=70) within the clade of arthropod lamins (*Figure 3* and *Figure 3—figure supplements 1* and *2*); it had previously been interpreted (*Mencarelli et al., 2011*) as closely related to cytoplasmic IF proteins of nematodes and therefore as an ortholog of the bilaterian cytoplasmic IF proteins, likely due to the narrower dataset used for their phylogenetic analysis. These results clearly challenge the identity of isomin as

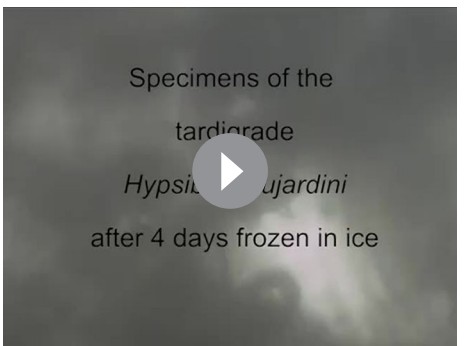

**Video 1.** The tardigrade *Hypsibius dujardini* survives freezing. This time-lapse video shows thawing specimens of the tardigrade *H. dujardini* after 4 days frozen in ice. One specimen starts with minuscule movements of one leg after 20 min of thawing and fully recovers locomotion within 120 min.

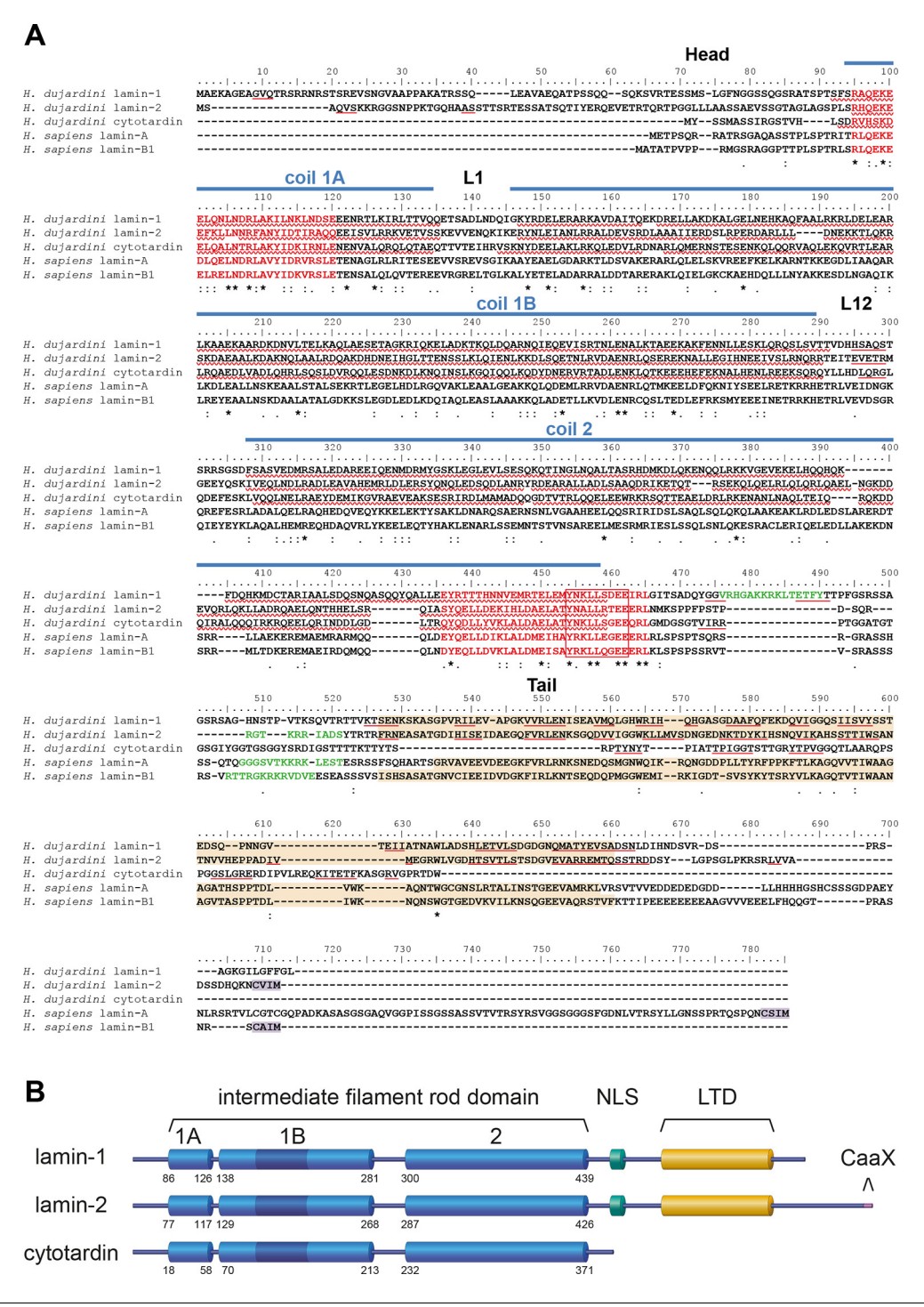

**Figure 2.** Structure and organization of the IF proteins of the tardigrade *Hypsibius dujardini*. (**A**) Protein sequence alignment of *H. dujardini* IF proteins with human (*Homo sapiens*) lamins A and B1. The names of *H. dujardini* IF proteins were chosen according their structural similarities to known IF proteins (lamin-1 and lamin-2: lamin-like; cytotardin: cytoplasmic IF-like; see text for details). Note the sequence similarities of the rod domains (coil 1A, L1, coil 1B, linker L12 and coil 2) and the intermediate filament consensus motifs (highlighted in red with highly conserved parts in a box) among all three proteins. The positions of rod sub-domains are placed as described for human IF proteins (review *Chernyatina et al., 2015*). The nuclear localization signal in lamin-1, lamin-2, lamin A and lamin B1 is highlighted in green and the immunoglobulin fold (Ig fold) is marked in light orange. Note the absence of an Ig fold in cytotardin. Predictions (Jpred3, JNetPRED) of α-helices and β-sheets are indicated by red

*Figure 2 continued on next page*

*Figure 2 continued*

waved underlines and solid underlines, respectively. The C-terminal prenylation motif of lamins (CaaX) is marked in purple. The alignment (Clustal Omega) has been performed using Analysis Tool Web Services from the EMBL-EBI (*McWilliam et al., 2013*). (*) indicates positions which have a single, fully conserved residue. (:) Indicates conservation between groups of strongly similar properties — scoring > 0.5 in the Gonnet PAM 250 matrix. (.) Indicates conservation between groups of weakly similar properties — scoring ≤ 0.5 in the Gonnet PAM 250 matrix. (B) Organization of the three IF proteins of *H. dujardini*. Dark blue colour in the 1B coil of the rod domain indicates six heptads that have been lost in the chordate lineage of cytoplasmic intermediate filament proteins. The numbers denote the amino acid positions of the beginning and end of each rod sub-domain. 1A, 1B and 2, coiled-coil segments of the rod domain; CaaX, isoprenylation motif at the carboxyl terminus; LTD, lamin tail domain; NLS, nuclear localization signal.

The following figure supplements are available for figure 2:

**Figure supplement 1.** Protein sequence alignment of *Hypsibius* dujardini cytotardin with selected human (*Homo sapiens*) cytoplasmic IF proteins.

**Figure supplement 2.** Protein sequence alignment of *Hypsibius* dujardini IF proteins with selected IF proteins from *Caenorhabditis elegans*.

---

a member of the bilaterian cytoplasmic IF protein clade (*Mencarelli et al., 2011*) and suggest that orthologs of genes encoding these proteins are entirely missing in arthropods, at least in those with known genomic sequences. In fact, besides the putative IF proteins from chelicerates, crustaceans and hexapods obtained from publicly available databases (e.g. GenBank), our transcriptomic and genomic analyses, which included screening of the genome of the centipede *Strigamia maritima* (see *Chipman et al., 2014*), the water flea *Daphnia pulex* (see *Colbourne et al., 2011*), and more than 70 transcriptomes from hexapod species sequenced as part of the 1KITE project (*Misof et al., 2014*), strongly suggest that these genes were already lost in the panarthropod lineage, since all of these panarthropod IF proteins cluster within a well-supported monophyletic clade of bilaterian lamins (GTR+G: BS=84, LG+G: BS=76; *Figure 3—figure supplements 1* and *2*). In this respect, even if the metazoan lamins are polyphyletic, as recently proposed by *Kollmar, 2015* based on the finding of putative nematocilin homologs in Bilateria, our results favour the tardigrade, copepod, and collembolan IF proteins as members of the bilaterian lamins rather than the bilaterian cytoplasmic IFs or nematocilins (*Figure 3—figure supplements 1* and *2*).

To determine their subcellular localization and organization, we generated antisera against the three IF proteins of *H. dujardini* and confirmed their specificity (see *Figure 4—figure supplement 1* and *Figure 5—figure supplement 1*). Immunolocalization of lamin-1, lamin-2 and cytotardin proteins in whole-mount preparations and cryosectioned specimens, in conjunction with confocal laser-scanning microscopy, revealed a highly specific subcellular distribution and tissue-restricted expression of these proteins in *H. dujardini*. As predicted, lamin-1 and lamin-2 proteins both display a typical, lamin-like distribution within all nuclei (*Figure 4A–D* and *Figure 4—figure supplement 2*). While lamin-1 is localised throughout the nucleoplasm (*Figure 4A*), lamin-2 is restricted to the nuclear periphery (*Figure 4B*), although there is a small overlap region between these two proteins (*Figure 4C*). The intranuclear distribution of lamin-1 corresponds to its lack of the CaaX motif, which is responsible for the association of lamins with the nuclear envelope (*Kitten and Nigg, 1991*). In contrast to the two lamins, cytotardin of *H. dujardini* is not localised within the nucleus, but in the peripheral cytoplasm of all epidermal and foregut cells, where it appears to be closely associated or aligned with the plasma membrane (*Figure 4E–G* and *Figure 4—figure supplement 3* and *Figure 6—figure supplement 1*). In this setting, it encircles the apical regions of the cells in a belt-like, filamentous array (*Figure 4E–G* and *Figure 4—figure supplement 3*).

In order to investigate the filament-forming capacity of cytotardin, we transiently transfected human mammary epithelial MCF-7 cells with a corresponding cDNA (*Figure 5A–F* and *Figure 5—figure supplement 1*). Immunofluorescence analyses using cytotardin antibody revealed that this protein is located exclusively in the cytoplasm of transfected MCF-7 cells, where it forms both short filaments and extensive cytoskeletal networks which most likely are homopolymeric (*Figure 5A–F*). Notably, some of the cytotardin arrays display cage-like perinuclear structures, while others are

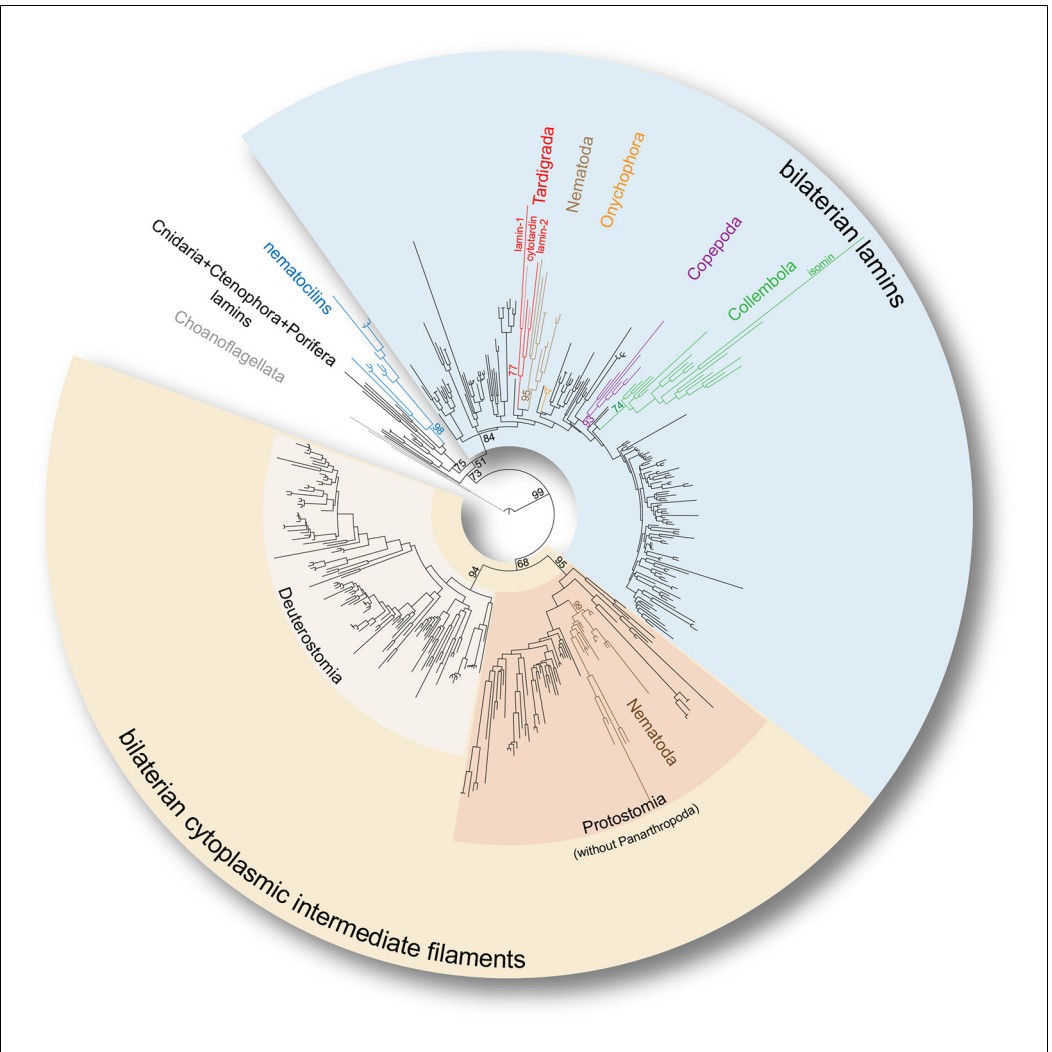

**Figure 3.** Phylogeny of the metazoan intermediate filament proteins illustrating the position of the three tardigrade IF proteins (highlighted in red). The tree was obtained from a Maximum likelihood analysis under a dataset-specific GTR+G substitution model of 447 eukaryotic intermediate filament proteins (see *Figure 3—figure supplement 1* for the full tree). Note that all tardigrade as well as collembolan (green) and copepod IF proteins (purple) belong to the bilaterian lamin clade (light blue). Hence, cytotardin and isomin are closer related to, for example, nematode lamins (light brown) than to nematode cytoplasmic IF proteins (dark brown), which are orthologs of the bilaterian cytoplasmic IF proteins (yellow). Selected bootstrap support values are given at particular nodes.

The following figure supplements are available for figure 3:

**Figure supplement 1.** Maximum likelihood tree under a dataset-specific GTR+G substitution model and accession numbers of 447 eukaryotic intermediate filament proteins and the placement of the IF protein genes of the tardigrade *Hypsibius dujardini* (highlighted in red).

**Figure supplement 2.** Maximum likelihood tree under the empirical LG+G substitution model and accession numbers of 447 eukaryotic intermediate filament proteins and the placement of the IF protein genes of the tardigrade *Hypsibius dujardini* (highlighted in red).

located in the periphery close to the cell membrane (*Figure 5E,F*). Double labelling for cytotardin and desmoplakin, a desmosomal protein mediating membrane attachment of mammalian IF proteins (*Simpson et al., 2011*), shows that cytotardin occurs close to desmosomes but is not co-localised with desmoplakin (*Figure 5C,D*). To examine whether this arrangement was mediated by

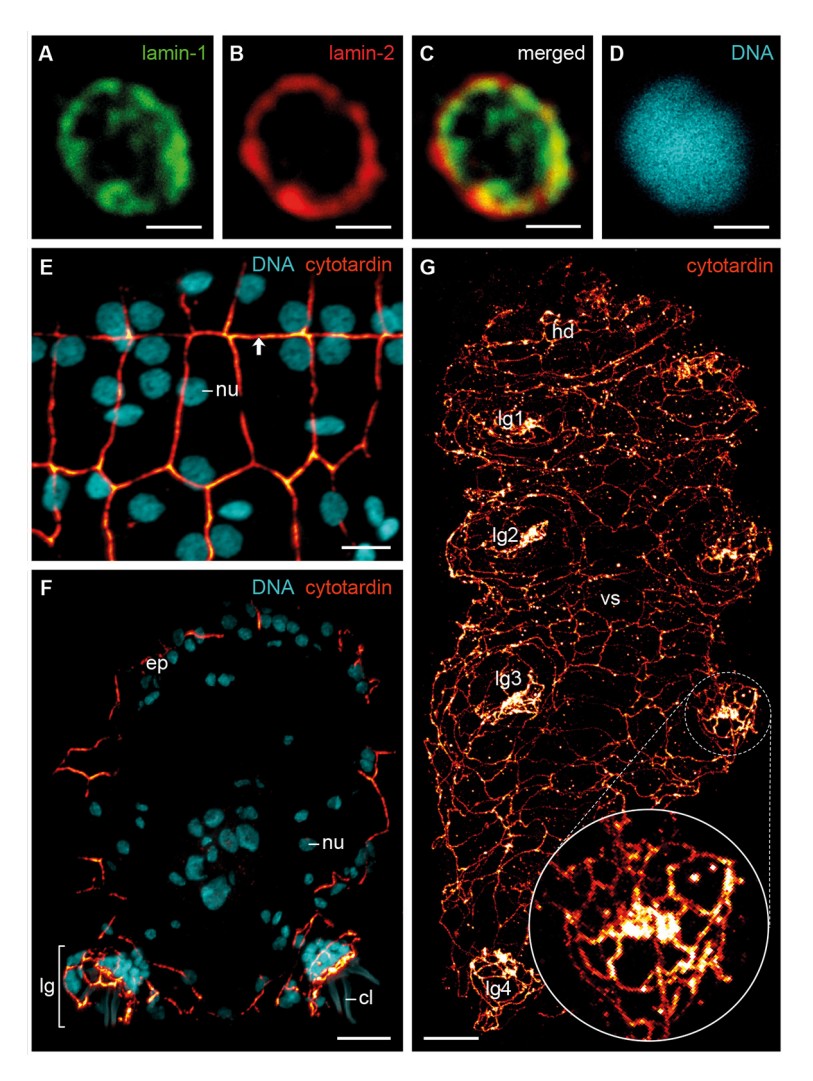

**Figure 4.** Immunofluorescence labelling of IF proteins in the tardigrade *Hypsibius dujardini*. Confocal laser-scanning micrographs. (**A–D**) Triple labelling of lamin-1 (green), lamin-2 (red) and DNA (cyan). Note the localization of lamin-1 within the nucleoplasm and that of lamin-2 at the nuclear periphery. (**E, F**) Double labelling of cytotardin (glow-mode) and DNA (cyan) on cryosections. (**E**) Tangential section of dorsolateral body wall. Arrow points to the dorsal midline. (**F**) Cross-section of a specimen. Dorsal is up. (**G**) Whole-mount preparation of a contracted specimen in ventral view. Anterior is up. Inset shows detail of the tip of a leg. cl, claw; ep, epidermis; hd, head; lg, leg; lg1–lg4, legs 1 to 4; nu, nucleus; vs, ventral body surface. Scale bars: (**A–D**) 1 µm, (**E**) 5 µm, (**F, G**) 10 µm.

The following figure supplements are available for figure 4:

**Figure supplement 1.** Western blots of lamin-1, lamin-2 and cytotardin antisera.

**Figure supplement 2.** Immunofluorescence labelling of lamin-1 and lamin-2 in the tardigrade *Hypsibius dujardini*.

**Figure supplement 3.** Immunofluorescence labelling of cytotardin in the tardigrade *Hypsibius dujardini* with focus on the epidermis.

interactions between cytotardin and keratins endogenously expressed in MCF-7 cells, we double-labelled the transfectants for cytotardin and keratin-8. Our data show that these two proteins are not co-localised and that the endogenous keratin networks are displaced in cells with dense cytotardin arrays (*Figure 5E,F*). These findings strongly support an intrinsic ability of the cytotardin protein

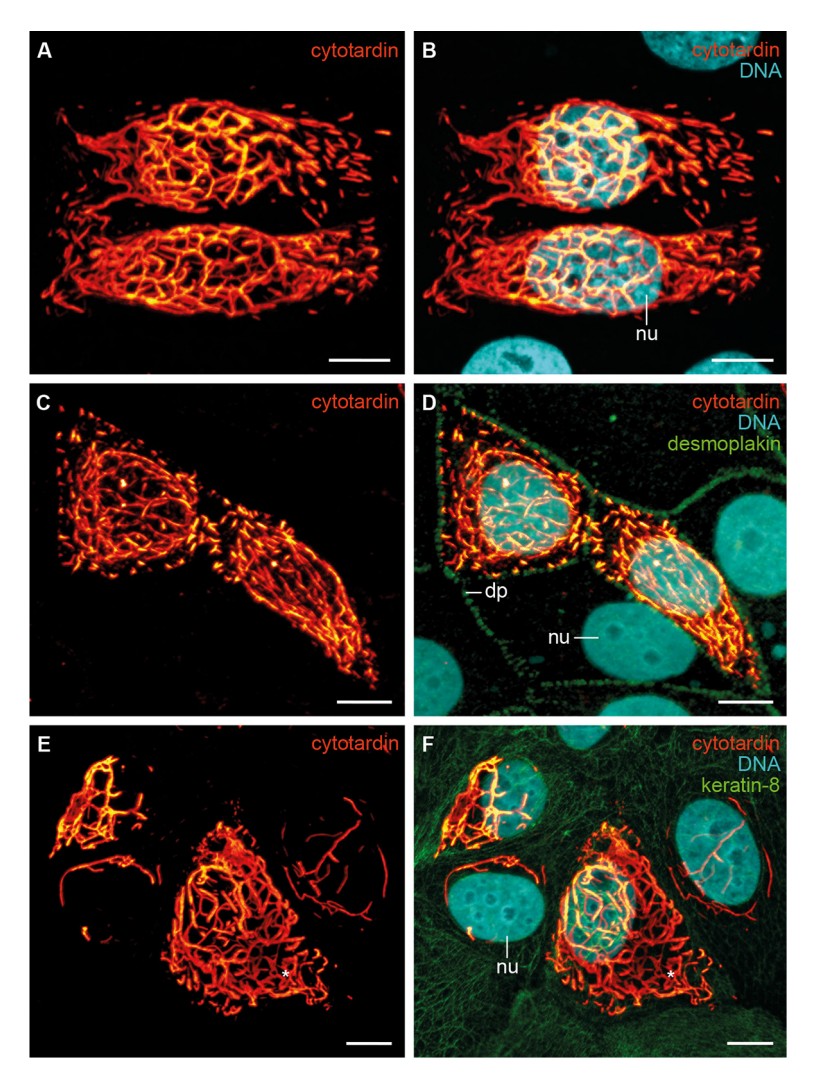

**Figure 5.** Immunolocalization of exogenous cytotardin in human MCF-7 epithelial cells. (**A**, **B**) Double labelling of cytotardin (glow-mode) and DNA (cyan). Note the cytoplasmic cytotardin filamentous network surrounding the nucleus and extensions close to cell borders. Short cytotardin filaments are aligned along the plasma membrane, different from the arrangement seen in tardigrade epithelial cells. (**C**, **D**) Triple labelling of cytotardin (glow-mode), desmoplakin (green) and DNA (cyan). Note that cytotardin forms a cytoplasmic filamentous network extending from the perinuclear area to the cell membrane. Note also that it is not co-localised with desmoplakin. (**E**, **F**) Triple labelling of cytotardin (glow-mode), keratin-8 (green) and DNA (cyan). Endogenous keratin networks have been displaced by cytotardin filaments from the perinuclear region without being disrupted (asterisk). dp, desmosomal plaque; nu, nucleus. Scale bars: (**A–F**) 10 μm.

The following figure supplements are available for figure 5:

**Figure supplement 1.** Characterization of cytotardin antiserum.

**Figure supplement 2.** Plasmids, primers, and restriction enzymes used for the cloning of tardigrade *lamin* and *cytotardin* genes.

of *H. dujardini* to both form homopolymeric filaments and cytoplasmic networks — both properties that are functionally analogous to mammalian cytoplasmic IFs (*Bohnekamp et al., 2015*).

Our data on tissue-specific distribution of cytotardin in *H. dujardini* further show that its belt-like arrays are confined to the ectodermal epithelia, including the epidermis, buccal tube, pharynx, and

oesophagus (*Figures 4E–G* and *6A,B* and *Figure 4—figure supplement 3* and *Figure 6—figure supplement 1*). Thus, the entire tardigrade body is ensheathed by a grid of belt-like filaments formed by the cytotardin protein, which retain their integrity even in contracted specimens (*Figure 4E–G* and *Figure 4—figure supplement 3* and *Figure 6—figure supplement 1*). The most prominent anti-cytotardin immunoreactivity is found in areas exposed to considerable physical stress in living specimens, including the bases of claws and the stylet apparatus (*Figures 4F,G* and *6B* and *Figure 4—figure supplement 3* and *Figure 6—figure supplement 1*). We therefore anticipate that much of the resistance of the tardigrade body to extreme conditions, such as cryobiosis (*Wright, 2001*; *Hengherr et al., 2009*), might be attributable to the dense, fibre-like cytotardin meshwork. The belt-like structures encircling each epidermal cell might help to resist the shearing

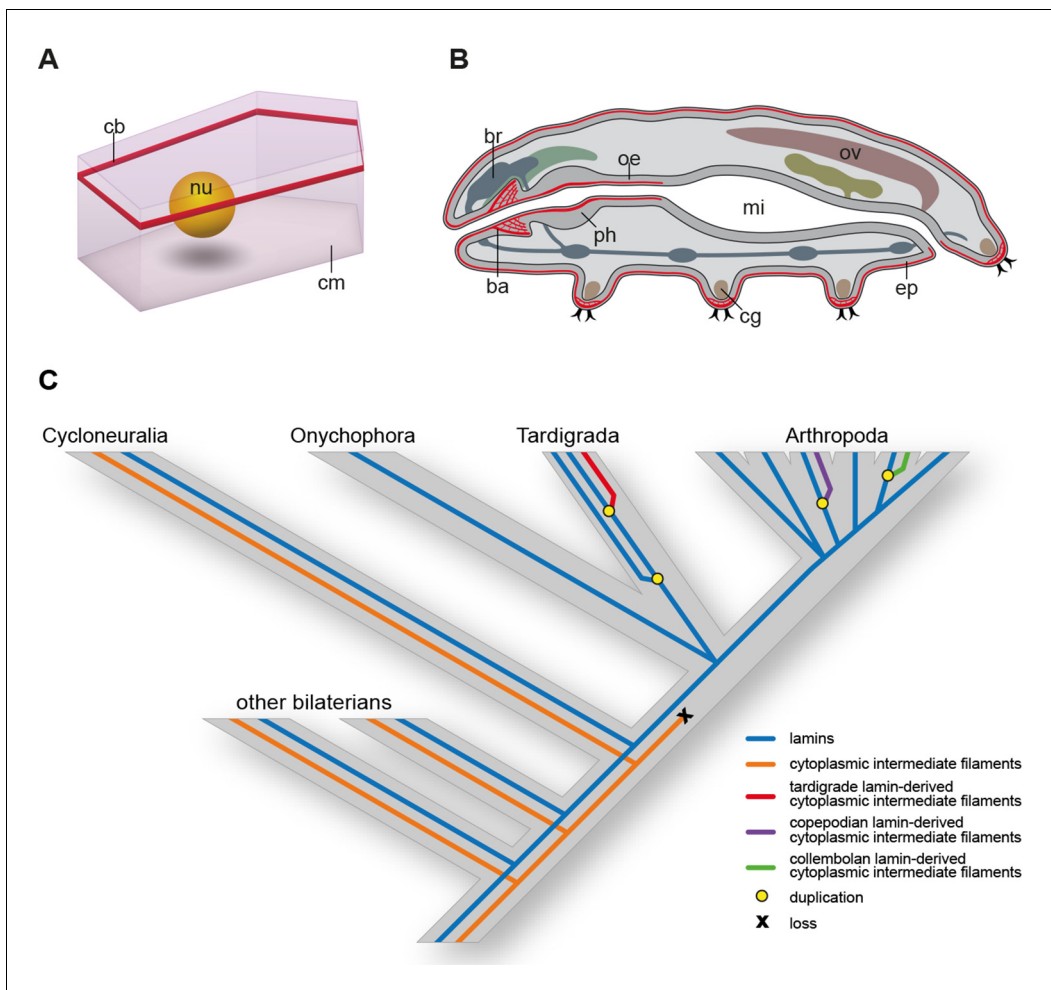

**Figure 6.** Distribution of cytotardin within the cell and across the tissues in *Hypsibius dujardini* and the evolutionary history of cytoplasmic intermediate filament proteins. (**A**) Diagram of an epidermal cell of *H. dujardini* with a belt-like arrangement of cytotardin. (**B**) Diagram of *H. dujardini* showing the distribution of cytotardin (red), which is confined to the ectodermal tissues. (**C**) Scenario of the independent origin of lamin-derived cytoplasmic intermediate filaments in tardigrades, collembolans, and copepods. Note that the cytoplasmic IFs in these three lineages (indicated in red, purple, and green, respectively) evolved independently from the cytoplasmic IFs of other bilaterians (highlighted in orange). ba, buccal apparatus; br, brain; cb, cytotardin filament belt; cg, claw gland; cm, cell membrane; ep, epidermis; mi, midgut; nu, nucleus; oe, oesophagus; ov, ovary; ph, pharynx.

The following figure supplement is available for figure 6:

**Figure supplement 1.** Immunofluorescence labelling of cytotardin in the tardigrade *Hypsibius dujardini* with focus on the foregut.

forces that arise during freezing and thawing cycles, whereas the dense meshwork at the basis of each claw and around the stylets might provide the tissue stability necessary for locomotion and feeding.

Together, our data demonstrate that cytotardin of *H. dujardini* is a cytoplasmic IF protein, predominantly expressed in ectodermal epithelia, that has evolved independently from the cytoplasmic IF proteins of other bilaterians by gene duplication and subsequent neofunctionalization (*Figure 6C*). This process was most likely triggered by an initial loss of the CaaX motif and nuclear localization signal from the ancestral *lamin* gene in the tardigrade lineage. Expression of the cytotardin protein in tissues that are subject to mechanical stress might have served as a pre-adaptation for the ability of tardigrades to survive extreme environmental conditions. Similar duplication and neofunctionalization events might have occurred in the collembolan and copepod lineages, as our findings demonstrate that these two taxa also show duplicated lamins that have lost their nuclear localization signals (*Figures 3* and *6C* and *Figure 3—figure supplements 1* and *2*). While isomin might stabilize the intestinal epithelial cells in collembolans (*Mencarelli et al., 2011*), the localization and function of the putative lamin-derived cytoplasmic IF protein in copepods has yet to be clarified.

## Materials and methods

### Culturing of specimens, transcriptomics, genomics and phylogenetic analyses

Specimens of *Hypsibius dujardini* (Doyère, 1840) were purchased from Sciento (Manchester, UK) and cultured as described by *Mayer et al., 2015*. The Illumina-sequenced transcriptomes of *H. dujardini* and five onychophoran species from *Hering et al., 2012* and *Hering and Mayer, 2014* were screened for expressed intermediate filament genes, including lamins, by BLAST searches (*Altschul et al., 1997*) with known metazoan lamins and cytoplasmic IF genes as bait sequences. Three putative IF protein genes from *H. dujardini* and one from each of the onychophoran species studied were verified afterwards by reciprocal BLAST searches against the nr database of GenBank (http://blast.ncbi.nlm.nih.gov/Blast.cgi). In addition, BLAST searches of the putative tardigrade intermediate filament genes in the genome of *H. dujardini* (http://badger.bio.ed.ac.uk/H_dujardini/; *Koutsovoulos et al., 2015*) yielded nearly identical sequences as obtained in our transcriptomic data, although the automatically predicted transcripts from the genome seem to be erroneous as previously reported by *Hering and Mayer, 2014* and had to be corrected manually. Furthermore, publicly available resources and databases [nr, TSA and EST databases of GenBank, Compagen (*Hemmrich and Bosch, 2008*)] as well as the genome of the centipede *Strigamia maritima* (see *Chipman et al., 2014*), the water flea *Daphnia pulex* (see *Colbourne et al., 2011*), and more than 70 transcriptomes from hexapod species sequenced as part of the 1KITE project (*Misof et al., 2014*) were comprehensively screened for putative eukaryotic intermediate filament genes. In total, 447 eukaryotic lamin and cytoplasmic IF genes were manually curated and selected for further analyses. Each of the sequences was verified by screening for the presence of a coiled-coil-forming domain (Pfam PF00038), a typical feature of intermediate filament proteins, and the presence or absence of a lamin tail domain (LTD; Pfam PF00932, SCOP d1ifra_, PDB 2lll/2kpw/1ufg/1ivt) using the webserver of Pfam 27.0 (*Finn et al., 2014*) and SMART (*Schultz et al., 1998*; *Letunic et al., 2015*), respectively. In addition, the occurrence of a nuclear localization signal (NLS motif) was predicted for all sequences by using the webserver of NucPred (*Brameier et al., 2007*) and cNLS Mapper (*Kosugi et al., 2009*) (cut-off score = 2.0). The protein structures (α-helices and β-sheets) were predicted for the tardigrade lamin-1, lamin-2 and cytotardin using Jpred3 and JNetPRED (*Cole et al., 2008*). For phylogenetic analyses, the rod domains of all sequences were aligned using the Mafft online version v7.245 (*Katoh and Standley, 2013*) with the most accurate option L-INS-i and default parameters. To remove homoplastic and random-like positions, the alignment was afterwards masked with the software Noisy rel. 1.15.12 (*Dress et al., 2008*) (-cutoff=0.8, -seqtype=P, -shuffles=20,000). Two Maximum likelihood analyses were conducted with the Pthreads version of RAxML v8.1.15 (*Stamatakis, 2014*). For each run, the best tree was obtained from 10 independent inferences and GAMMA correction of the final tree under either the empirical LG substitution model or a dataset-specific GTR substitution matrix. The LG model was automatically selected by RAxML (PROTGAMMAAUTO option) as best-fitting substitution model, which is in line with the best-fitting

model obtained with ProtTest v3.4.1 (*Darriba et al., 2011*) (LG+G) according to the Akaike information criterion (*Akaike, 1974*) (AIC), Bayesian Information Criterion (*Schwarz, 1978*) (BIC), corrected AIC (*Sugiura, 1978*; *Hurvich and Tsai, 1989*), and Decision Theory Criterion (*Minin et al., 2003*) (DT). Notably, the analysis using the dataset-specific GTR+G model yielded a better log likelihood score (−174,216.74) for the best tree than using the best obtained empirical model LG+G (−178,092.26). Bootstrap support values for both trees were calculated using the rapid bootstrapping algorithm implemented in RAxML from 1,000 pseudoreplicates. The protein domain structures of the sequences analysed were mapped on the trees using iTol v2 (*Letunic and Bork, 2011*).

## Cloning

Total RNA from several hundred specimens of *H. dujardini* was extracted and purified using TRIzol Reagent (Life Technologies, Carlsbad, CA) and RNeasy MinElute Cleanup Kit (Qiagen, Hilden, Germany) according to the manufacturers' protocols. First strand cDNA synthesis was performed using random hexamer primer and SuperScript II Reverse Transcriptase (Life Technologies) and afterwards used as template for amplification of the whole coding sequence (CDS) of *lamin-1, lamin-2* and *cytotardin* using gene specific primers. The *cytotardin* specific primers contained restriction sites that were required for subsequent cloning into bacterial or mammalian expression vectors (see *Figure 5—figure supplement 2*). The amplicons of *lamin-1* and *lamin-2* were cloned into the pGEM-T Vector System (Promega, Madison, WI) to generate the plasmids pGEM-T-lamin-1 and pGEM-T-lamin-2. *Cytotardin* CDS was cloned into expression vectors pET15b (Novagen Merck Millipore, Darmstadt, Germany), pcDNA3.1/Zeo $^{(+)}$ (Life Technologies) and pEGFP-C3 (Clontech Laboratories, Mountain View, CA) to generate the following plasmids: pET15b-cytotardin, pcDNA3-cytotardin, pcDNA3-HA-cytotardin and pEGFP-cytotardin. All PCR-amplified constructs were verified by Sanger sequencing and have been deposited in GenBank (http://www.ncbi. nlm.nih.gov/genbank) under accession numbers KU295460–KU295467.

## Antibody generation

Polyclonal antibodies against HPLC-purified synthetic peptides (C-terminal) of lamin-1, lamin-2 and cytotardin of *H. dujardini* were newly generated, following coupling to KLH (key limpet hemocyanin) (Peptide Specialty Laboratories GmbH, Heidelberg, Germany). Anti-lamin-1 (antigen: SNLDIHNDSV-RDSPRSAG-C) and anti-cytotardin (antigen: EQKITETFKASGRVGPRTDW-C) were purified from sera of immunised guinea pigs and anti-lamin-2 (antigen: REMTQSSTRDDSYLGPSGLPKR-C) from the serum of immunised rabbits (Peptide Specialty Laboratories GmbH).

## Immunolocalization in whole-mount preparations and cryosections

For cryosectioning, specimens of *H. dujardini* were concentrated by filtering the culture medium through a polyamide mesh (pore size: 30 µm). The concentrated specimens were transferred into an embedding medium for cryosectioning (Tissue-Tek O.C.T Compound; Sakura Finetek, Staufen, Germany). Single drops of the medium containing the tardigrades were immediately frozen in dry ice-cooled 2-methylbutane. The frozen drops were cryosectioned into 5 µm thick sections. The sections were attached to SuperFrost Plus slides (Menzel, Braunschweig, Germany), then dried at room temperature and stored at −80°C. The ice-cooled cryosections were fixed on slides with a 4% solution of formalin (FA) freshly prepared from paraformaldehyde in phosphate-buffered saline (PBS; 0.1 M, pH 7.4) for 15 min. The sections were then shortly washed with an ammonium chloride solution (50 mM in PBS) and rinsed in Tris-buffered saline (TBS; 0.01 M, pH 7.6) two times for 3 min each. The anti-lamin-1 serum was applied to cryosections at a concentration of 1.9 µg/mL in TBS containing 1% bovine serum albumin (BSA). Equally applied were the anti-lamin-2 serum with a concentration of 1.8 µg/mL and the anti-cytotardin serum with a concentration of 2.1 µg/mL. The incubation with the primary antibody solution was performed at room temperature for one hour. After two 3-min washing steps in TBS, a secondary antibody, either donkey anti-guinea pig Alexa Fluor 488 (3 µg/mL in TBS with 1% BSA; Jackson ImmunoResearch Laboratories, Hamburg, Germany) for anti-lamin-1 and anti-cytotardin or goat anti-rabbit Alexa Fluor 568 (4 µg/mL in TBS with 1% BSA; Molecular Probes, Darmstadt, Germany) for anti-lamin-2, was applied at room temperature for one hour. The sections were then washed in TBS two times for 3 min and shortly rinsed with distilled water and afterwards with ethanol (100%).

For whole-mount immunohistochemistry, specimens were first concentrated as described above and then pipetted into a 1 mL Eppendorf tube. After carefully removing excess water, the specimens were flash-frozen by placing each tube on dry ice. Frozen specimens were fixed immediately by applying a 4% solution of formaldehyde (FA) in PBS with 1% dimethyl sulfoxide (DMSO). The fixative was applied at room temperature overnight. The whole-mounts were then washed in PBS two times for 10 min, two times for 30 min and two times for an hour. The specimens were cleared by dehydration in an ascending ethanol series (70%, 90%, 95%, 100%, 100% ), applying xylene as a clearing agent two times for 3 min and rehydration in a descending ethanol series (100%, 100%, 90%, 70%, 50% ). The whole-mount preparations were then washed at 37°C in PBS two times for 10 min. A mixture of collagenase/dispase (each 1 mg/mL; Roche Diagnostics, Mannheim, Germany) and hyaluronidase (1 mg/mL; Sigma-Aldrich, Munich, Germany) diluted in PBS was applied at 37°C for 10 min. After the incubation with enzymes, a 15-min post-fixation with 4% FA in PBS followed at room temperature. The specimens were washed in an ammonium chloride solution (50 mM) two times for 15 min, afterwards in PBS with 1% Triton X-100 (Sigma-Aldrich) two times for 10 min, two times for 30 min, once for an hour, overnight and then two times for 1 min. The whole-mount preparations were incubated in a blocking solution containing 10% normal goat serum (NGS; Sigma-Aldrich), 1% Triton X-100 and 1% DMSO in PBS at room temperature for one hour. The anti-cytotardin serum with a concentration of 4.2 µg/mL in PBS with 1% NGS, 1% DMSO and 0.02% sodium azide was applied at room temperature for three days. After washing in PBS with 1% Triton X-100 and 1% DMSO three times for 5 min, two times for 15 min, four times for two hours, overnight and two times for 15 min a secondary antibody (donkey anti-guinea pig Alexa Fluor 488; Jackson ImmunoResearch Laboratories) with a concentration of 3 µg/mL in PBS with 1% NGS, 1% DMSO and 0.02% sodium azide was applied at room temperature for three days. Finally the specimens were washed in PBS with 1% Triton X-100 and 1% DMSO three times for 5 min, three times for 15 min and two times for one hour and afterwards in PBS (without Triton X-100 and DMSO) four times for 15 min.

Either DAPI (1 µg/mL; Carl Roth, Karlsruhe, Germany), SYBRGreen I (diluted 1:10,000; Molecular Probes), propidium iodide (0.5 µg/mL; Carl Roth) or TO-PRO-3 iodide (diluted 1:1,000; Molecular Probes) were used as nuclear counterstain markers. Each of these fluorescent dyes was simply added to the solution containing the secondary antibody. Cryosections and whole-mount preparations were mounted in Prolong Gold antifade reagent (Molecular Probes). The whole-mount preparations were mounted between two cover slips using petrolatum at all corners as spacer. All slides and cover slips were sealed at the edges with transparent nail polish for long time storage. Specimens were analysed with the confocal laser-scanning microscopes Zeiss LSM 780 (Carl Zeiss Microscopy, Göttingen, Germany) and Leica TCS STED (Leica Microsystems CMS, Wetzlar, Germany).

## Recombinant protein expression of cytotardin in bacteria

*Escherichia coli* BL21 (DE3) were transformed with pET15b or pET15b-cytotardin vectors. Expression of recombinant protein was induced with 0.1 mM IPTG for three hours in *Luria Bertani* (LB) medium. Bacterial suspensions were centrifuged at 3,000 g for 10 min at 4°C. Bacterial pellets were then washed twice in PBS and lysed in 5x Laemmli buffer (Volume in µL = OD600/10xVolume of culture in mL) containing protease inhibitor (#78439; Thermo Scientific, Waltham, MA), heated at 95°C for 10 min, briefly sonicated and heated again. Lysate of cells transformed with pET15b-cytotardin was diluted at 1:50. Recombinant cytotardin protein expression was first assessed by Coomassie blue stained SDS-PAGE and further by Western blotting.

## Transfection and immunolabelling of cytotardin in MCF-7 cell culture

Cells from the human cell line MCF-7 (ATCC HTB-22) were cultured in Dulbecco's modified Eagle's medium (DMEM, GE Healthcare, Little Chalfont, UK) supplemented with 10% foetal bovine serum (FBS, GE Healthcare), 100 U/mL penicillin, and 100 µg/mL streptomycin. Cells were transfected with pcDNA-cytotardin, pcDNA-HA-cytotardin, pEGFP-cytotardin and the corresponding empty vectors using Xfect Transfection reagent (Clontech Laboratories) according to the manufacturer's protocol. For immunofluorescence microscopy, cells were grown on glass slides and fixed 24 hr after transfection. MCF-7 cells were washed twice with PBS and fixed either in methanol or in 4% FA freshly prepared from paraformaldehyde in PBS. For methanol fixation, cells grown on a coverslip were put in ice-cold methanol for 10 min, acetone for 1 min and dried at room temperature for 30 min. For FA

fixation, cells were put in a 4% FA solution in PBS for 15 min at room temperature. After fixation, cells were washed in PBS and used for immunolabelling. FA-fixed cells were first permeabilised in PBS 0.1% Triton X-100 for 5 min and blocked in PBS 5% bovine serum albumin for 30 min. Methanol- and FA-fixed cells were then incubated overnight at 4°C with primary antibody solutions, washed three times, incubated in secondary antibodies + DAPI (Sigma-Aldrich) solution for one hour, washed two times in PBS, once in water, once in 100% ethanol, dried for 30 min and mounted in mounting medium (Dianova, Hamburg, Germany). Primary antibodies were used at the following dilutions: guinea pig anti-cytotardin, 1:1,000; mouse anti-desmoplakin (11-5F), 1:150; mouse anti-keratin-8 (Ks 8.7), 1:100; mouse anti-HA-tag (MMS-101P; Covance, Princeton, NJ), 1:100. Images were taken using a confocal laser-scanning microscope (LSM 780; Carl Zeiss Microscopy) with 63×/1.46 NA oil immersion objective and Z-stack images were assembled by 'Maximum Intensity Projection' of the ZEN 2012 software (Carl Zeiss Microscopy).

## Western blotting and specificity tests

Western blots were performed either on lysates of transfected MCF-7 cells or multiple specimens of *H. dujardini*. Transfected MCF-7 cells grown in 10 cm$^2$ dishes were lysed in 200 µL 5x Laemmli sample buffer, boiled for 10 min at 98°C and briefly centrifuged prior to use. Concentrated specimens of *H. dujardini* were lysed in 800 µL 5x Laemmli buffer containing protease inhibitor, boiled for 10 min, sonicated and boiled again for 10 min. SDS-PAGE and Western blotting were performed as described previously (*Löffek et al., 2010*). Primary antibodies, diluted in Tris-buffered saline containing 0.05% Tween 20 (A4974; AppliChem, Darmstadt, Germany), were used at the following concentrations: anti-lamin-1, 1:10,000; anti-lamin-2, 1:5,000; anti-cytotardin, 1:10,000, anti-HA tag (MMS-101P; Covance), 1:1,000 and anti-GFP (sc-5385; Santa Cruz Biotechnology, Dallas, TX), 1:1,000. Anti-lamin-1 and anti-lamin-2 antibodies were only tested in Western blots based on lysates of tardigrade specimens.

## Acknowledgements

We thank Sandra Treffkorn and Vladimir Gross for help with tardigrade culture, Fanny Loschke for assistance with confocal microscopy, Susann Kauschke for the light micrograph in *Figure 1*, Miriam Richter and Gabi Baumbach for technical assistance with cloning and cell culture, and Vladimir Gross, Jens Bohnekamp and Matthias Behr for critical discussions. We are grateful to members of Luca Borradori's laboratory (Bern, Switzerland) for providing the pet15b vector. This work was partially supported by the German Research Foundation (DFG; INF 268/230-1; MA1316/21-1) to TMM and the Emmy Noether Programme of the DFG (Ma 4147/3-1) to GM. The authors declare no conflicts of interests.

## Additional information

### Funding

| Funder | Grant reference number | Author |
|---|---|---|
| Deutsche Forschungsgemeinschaft | INF 268/230-1 | Thomas M Magin |
| Deutsche Forschungsgemeinschaft | MA1316/21-1 | Thomas M Magin |
| Deutsche Forschungsgemeinschaft | Ma 4147/3-1 | Georg Mayer |

The funders had no role in study design, data collection and interpretation, or the decision to submit the work for publication.

### Author contributions

LH, J-EB, JR, Acquisition of data, Analysis and interpretation of data, Drafting or revising the article; TMM, GM, Conception and design, Analysis and interpretation of data, Drafting or revising the article

**Author ORCIDs**

Lars Hering, http://orcid.org/0000-0003-4469-991X

## Additional files

### Major datasets

The following datasets were generated:

| Author(s) | Year | Dataset title | Dataset URL | Database, license, and accessibility information |
|---|---|---|---|---|
| Hering L, Bouameur J-E, Reichelt J, Magin TM, Mayer G | 2016 | Hypsibius dujardini cytotardin | http://www.ncbi.nlm.nih.gov/nuccore/KU295462 | Publicly available at the NCBI GenBank (accession no. KU295462). |
| Hering L, Bouameur J-E, Reichelt J, Magin TM, Mayer G | 2016 | Hypsibius dujardini lamin-1 | http://www.ncbi.nlm.nih.gov/nuccore/KU295460 | Publicly available at the NCBI GenBank (accession no. KU295460). |
| Hering L, Bouameur J-E, Reichelt J, Magin TM, Mayer G | 2016 | Hypsibius dujardini lamin-2 | http://www.ncbi.nlm.nih.gov/nuccore/KU295461 | Publicly available at the NCBI GenBank (accession no. KU295461). |
| Hering L, Bouameur J-E, Reichelt J, Magin TM, Mayer G | 2016 | Euperipatoides rowelli lamin | http://www.ncbi.nlm.nih.gov/nuccore/KU295463 | Publicly available at the NCBI GenBank (accession no. KU295463). |
| Hering L, Bouameur J-E, Reichelt J, Magin TM, Mayer G | 2016 | Phallocephale tallagandensis lamin | http://www.ncbi.nlm.nih.gov/nuccore/KU295464 | Publicly available at the NCBI GenBank (accession no. KU295464). |
| Hering L, Bouameur J-E, Reichelt J, Magin TM, Mayer G | 2016 | Ooperipatus hispidus lamin | http://www.ncbi.nlm.nih.gov/nuccore/KU295465 | Publicly available at the NCBI GenBank (accession no. KU295465). |
| Hering L, Bouameur J-E, Reichelt J, Magin TM, Mayer G | 2016 | Principapillatus hitoyensis lamin | http://www.ncbi.nlm.nih.gov/nuccore/KU295466 | Publicly available at the NCBI GenBank (accession no. KU295466). |
| Hering L, Bouameur J-E, Reichelt J, Magin TM, Mayer G | 2016 | Eoperipatus sp. LH-2012 lamin | http://www.ncbi.nlm.nih.gov/nuccore/KU295467 | Publicly available at the NCBI GenBank (accession no. KU295467). |

The following previously published datasets were used:

| Author(s) | Year | Dataset title | Dataset URL | Database, license, and accessibility information |
|---|---|---|---|---|
| Koutsovoulos G, Kumar S, Laetsch DR, Stevens L, Daub J, Conlon C, Marhoon H, Thomas F, Aboobaker A, Blaxter M | 2015 | Data from: The genome of the tardigrade Hypsibius dujardini (doi: 10.1101/033464) | http://badger.bio.ed.ac.uk/H_dujardini/ | Publicly available at http://badger.bio.ed.ac.uk/H_dujardini/ |

| | | | | |
|---|---|---|---|---|
| Boothby TC, Tenlen JR, Smith FW, Wang JR, Patanella KA, Osborne Nishimura E, Tintori SC, Li Q, Jones CD, Yandell M, Messina DN, Glasscock J, Goldstein B | 2015 | Data from: Evidence for extensive horizontal gene transfer from the draft genome of a tardigrade (doi: 10.1073/pnas.1510461112) | http://weatherby.genetics.utah.edu/seq_transf/ | Publicly available at http://weatherby.genetics.utah.edu/seq_transf/ |
| Chipman AD, et al. | 2014 | Data from: The first myriapod genome sequence reveals conservative arthropod gene content and genome organisation in the centipede Strigamia maritima (doi: 10.1371/journal.pbio.1002005) | http://metazoa.ensembl.org/Strigamia_maritima | Publicly available at http://metazoa.ensembl.org/Strigamia_maritima |
| Adamski M, Leininger S, Bergum B, Liu J, Adamska K | 2013 | Protein translations from Sycon ciliatum transcriptome (SCIL_T-PEP_130802) | http://www.compagen.org/datasets/SCIL_T-PEP_130802 | Publicly available at the Compagen platform for early branching metazoan animals (http://www.compagen.org/datasets.html). |
| Richter DJ, Mora J, Nichols SA | 2013 | Protein translations from Oscarella carmela transcriptome (OCAR_T-PEP_130911) | http://www.compagen.org/datasets/OCAR_T-PEP_130911 | Publicly available at the Compagen platform for early branching metazoan animals (http://www.compagen.org/datasets.html). |
| | 2012 | Oscarella sp. transcriptome (Osp._T-CDS_120614) | http://www.compagen.org/datasets/Osp._T-CDS_120614 | Publicly available at the Compagen platform for early branching metazoan animals (http://www.compagen.org/datasets.html). |
| Lapébie P, Ruggiero A, Chevalier S, Chang P, Momose T | 2014 | Data from: Differential Responses to Wnt and PCP Disruption Predict Expression and Developmental Function of Conserved and Novel Genes in a Cnidarian (doi: 10.1371/journal.pgen.1004590) | http://www.compagen.org/datasets/CHEM_T-CDS_141022 | Publicly available at the Compagen platform for early branching metazoan animals (http://www.compagen.org/datasets.html). |
| Moroz LL, Williams PL, Kohn A | 2013 | Bolinopsis infundibulum transcriptome (Bolinopsis_infundibulum_Illumina_RNA-Seq) | http://neurobase.rc.ufl.edu/pleurobrachia/browse | Publicly available at NeuroBase: A Comparative Neurogenomics Database (http://neurobase.rc.ufl.edu/pleurobrachia/browse). |
| Moroz LL, Williams PL, Kohn A | 2015 | Beroe abyssicola transcriptome (Beroe_abyssicola_Illumina_RNA-Seq) | http://neurobase.rc.ufl.edu/pleurobrachia/browse | Publicly available at NeuroBase: A Comparative Neurogenomics Database (http://neurobase.rc.ufl.edu/pleurobrachia/browse). |

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
