## [Decision Letter]

Thank you for submitting your work entitled "Novel origin of lamin-derived cytoplasmic intermediate filaments in tardigrades" for consideration by *eLife*. Your article has been reviewed by four peer reviewers, and the evaluation has been overseen by a Guest Reviewing Editor (who also served as one of the reviewers) and Randy Schekman as the Senior Editor.

The following individuals involved in review of your submission have agreed to reveal their identity: Michael Schaffeld (peer reviewer 2) and Reimer Stick (peer reviewer 3).

The reviewers have discussed the reviews with one another and the Reviewing editor has drafted this decision to help you prepare a revised submission.

General assessment:

All four reviewers found the article very interesting, i.e. intermediate filament (IF) proteins in evolution. Here is a comprehensive phylogenetic analysis about their appearance in the water bear (Tardigrada) as an example of a very simple and fascinating "micro-animal" and how their sequences fit into the complex evolutionary tree of IF proteins.

Central conclusions:

1) The water bear *Hypsibius dujardini* genome exhibits three coding units for potential intermediate filament (IF) proteins, two with a nuclear localization signal sequence, one without.

2) A potential structural fold is introduced and the proteins are grouped – according to their primary sequence – into an evolutionary tree.

3) The proteins are called, because of sequence homology with lamins, lamin-1 to -3. Lamin-1 and lamin-2 exhibit nuclear localization signals and an Ig fold-like domain. Lamin-2 does not have a CAAX box for isoprenylation and hence resembles a lamin C of vertebrates.

4) Lamin-1 and lamin-2 are localized at the nuclear envelope in authentic *Hypsibius dujardini* tissue sections, lamin-3 is found in epithelial cells. It is essential to change the name lamin-3, since naming a cytoplasmic protein "lamin" will confuse readers tremendously. For instance, Hd-IF-1 would make sense – provided the protein is demonstrated to be really IF-like by sequence criteria.

5) Transfection of lamin-3 cDNA into human epithelial MCF-7 cells revealed that these proteins were able to form filaments that organized into extensive cytoplasmic networks.

6) The cytoplasmic IF protein of *Hypsibius dujardini* is most probably, like the cytoplasmic IF protein of *Caenorhabditis elegans*, derived from the nuclear version by loss of the nuclear localization signal.

Required revisions:

The reviewers raise a number of concerns that should be appropriately considered before the paper can be accepted. Those comments concerning the sequences used to construct the "tree" may cause some extra work, but the remainder of the comments concern the improvement of the text, in particular the background of the research.

1) The amino acid sequence of the α-helical rod provides a good starting point to characterize potential intermediate filament (IF) proteins by comparing them to authentic vertebrate ones. As the conserved end domains are evolutionary signatures with functional roles, those segments should be looked at in particular. Hence, an amino acid comparison of lamin-1 to lamin-2 to lamin-3 with each other and with e.g. human, *C. elegans* and Hydra or a protist lamin would be a meaningful way to analyze these new proteins before drawing domain schemes.

2) With respect to the last comment, the work by Martin Kollmar (Scientific Reports 5:10652) has to be discussed appropriately.

3) The fact that cytoplasmic IF proteins (which are essentially not lamins) are derived from lamins is not so new as one would learn from the work over the last two decades of the Weber group in Göttingen. How can one leave out this worm in light of its close relatedness to water bears?

4) A figure with an evolutionary "tree" that can be printed on one page would be good – as it would only allow giving main relationships. The Weber group did such overviews in their papers.

5) The issue of cytoplasmic IFs – as well as isomin – may be discussed with a broader perspective. In the light of the million insect species, *Drosophila* is surely not "the" insect.

6) The sequences used to build the tree should be "cleaned".

7) IF proteins form – with very rare exceptions (Keratin 18) – parallel coiled-coil dimers. Meaningful molecular modeling should consider that fact.

Full Reviews

Reviewer #1:

Lars Hering and colleagues report on the occurrence of both nuclear and cytoplasmic intermediate filament (IF) proteins in Tardigrades, the water bear. Based on in silico work done for *Hypsibius dujardini* they derive two amino acid sequences that exhibit similarity to lamins and one that is cytoplasmic. As tardigrades are very special organisms, tolerating various extreme stresses, they are of high interest for every life scientist. These data they use to investigate evolutionary relationships. The evolutionary trees provided are interesting, but the definition of the proteins is poor (see below). But immuno-cytochemical methods the authors demonstrate that the two nuclear IF proteins localize to the nuclear envelope. Moreover, these authors use cDNAs to transfect human epithelial cells and demonstrate that the cytoplasmic lamin-3 (what a misnomer) assembles into some type of regular network. All the cell biological data are very interesting and of high quality.

For a reader following the DNA-RNA-protein dogma for the last decades, a protein is still defined by its sequence, and not its position in a phylogenetic tree. Old fashioned as it may be, it would be good to discuss such issues as domain size, coiled-coil forming ability, i.e. the occurrence of heptad or hendcad patterns. The sequences are, however, not shown neither are they directly compared with authentic lamins such as those from Hydra, man or *C. elegans*. Strange enough, the latter organism is not centrally discussed, as for this worm Klaus Weber and his group in Göttingen demonstrated how lamins and invertebrate cytoplasmic intermediate filament proteins are related, i.e. by loss of the nuclear localization signal (NLS) the cytoplasmic IF proteins emerged (Dodemont et al. 1990 & 1994 EMBO J., Andreas Erber et al. 1998 & J. Mol. Evol., and several more papers). Obviously a fruitful strategy.

A selected paradigmatic-type comparison would be very instructive. I took the CAAX-box-free version and compared its sequence with human lamin A: 1. Coil 1B is missing 4 amino acids and coil 2 carries a 3-amino acid insertion. The size of these helical segments including coil 1A is highly conserved, with the rare exception of coil 2 of the single lamin of *C. elegans*, which is missing 2 heptads or 14 amino acids. Notably, the IF consensus motif in coil 1A exhibits high conservation as does the IF consensus motif at the end of the rod in coil 2. More surprisingly, the linker segment L1 is entirely kept both to the very amino acid number and with respect to the quality of amino acids, which all are good for α-helix formation; the same holds true for linker L12 from what one can conclude that structural principals in the α-helical rod domain are highly conserved. Figure 1 does not give any clue to how related these domains are. In Figure 1 the subdivision in 2A and 2B is completely illusory, it is wrong. The authors should refer to a recent structural review of Sergei Strelkov (Leuven) in Current Opinion in Cell Biology (2015). "Head" and "tail" sequences vary very much in IF proteins – notably here: the Ig fold sequence in the carboxyl-terminal domain exhibits highest identity with that of human lamin A, when the whole amino acid sequence is considered. The NLS shows an interesting variation from the standard, both CDK1 sequences flanking the rod are there but slightly shifted compared with the position of lamin A. In summary, Figure 1 gives the impression that everything is like in every IF protein and that lamin-1 to -3 are identical: to what extent are they? The tree in Figure 1 does not tell anything in detail. So how different is the tardigrade cytoplasmic IF protein from its lamins, and how closely is it related to human keratins K8 and K18? Does it have features that are anything like "keratin-related"? Along these lines: the structural models shown as Figure 1—figure supplement 3 are pretty meaningless – all three lamin-1 to -3 should form coiled coils.

Additional data files and statistical comments:

The amino acid sequences and the kind how coiled-coil forming regions are present and can/do define a domain should definitively be provided.

Reviewer #2:

By whole-transcriptome analysis (using NGS), database mining and subsequent thorough phylogenetic analysis Hering and colleagues revealed that the ancestral genes encoding cytoplasmic IFs in bilaterians have once been lost in the entire panarthropod lineage, including the soft skinned velvet worms (Onychophora) and water bears (Tardigrada). This important finding suggests that the development of an exoskeleton may not have been the main driving force for the loss of cytoplasmic IFs in euarthropods. Moreover, the presented phylogenetic analysis strongly supports that several members of this taxon, notably tardigrades, collembolans and copepods each independently evolved novel cytoplasmic IFs, again originating from gene duplications of a nuclear lamin gene, long after the first bilaterian cytoplasmic IF arose. Furthermore, the authors demonstrated the capacity of the novel cytoplasmic IF protein lamin-3 to form IF networks within transfected cultured cells and traced its expression in the tardigrade *H. dujardini*, showing a unique, belt-like distribution within the peripheral cytoplasm of ectodermally derived epithelial cells, such as the epidermis.

In the study presented here, the authors do not only provide a sound and comprehensive phylogenetic analysis, but also combine their results with a broad survey of expression data as well as functional analyses. This is a well and carefully reasoned scientific approach, providing important insights into the early evolutionary steps that ultimately led to the high diversity of metazoan IF proteins as well as to the role of selective pressure that may have triggered those evolutionary events. The structure and function of a given protein or protein family is a result of a long-term evolutionary process and therefore cannot be completely understood without such phylogenetic considerations. Overall the results and conclusions presented in the submitted manuscript are convincing, making a valuable contribution to both, a better understanding of the evolutionary history of the IF protein family and how the novel cytoplasmic IF may account for the tardigrades ability to withstand the extraordinary mechanical stress enforced by extreme environmental conditions such as desiccation and cryobiosis.

Therefore, this referee strongly recommends the publication of the submitted manuscript in *eLife*.

Reviewer #3:

Hering et al. report on the evolution of cytoplasmic intermediate filament proteins (cIF) in the panarthropod lineage. They have characterized three tardigrade IF proteins in more detail: two lamins and one cytoplasmic IF protein. They convincingly show by immunofluorescence microscopy that the two lamins are nuclear and that the third protein is cytoplasmic. It forms filamentous structures in the cytoplasm of epithelial cells. This part of the manuscript is clear and convincing.

From their data they conclude that the genes for cytoplasmic IF proteins have been lost at the base of the panarthropod lineage and that cIF genes have been acquired at least three times independently in collembolans, copepods, and tardigrades. This is a very interesting point. This statement is based solely on evolutionary tree construction of a broad range of IF rod sequences. However, it is not fully convincing. For the collembolan lineage it is based solely on the fact that the isomin rod sequence groups with bilaterian lamin rod sequences. At none of the nodes of the two trees bootstrap values exceed 50%. Given that collembolans are at the base of the arthropod lineage this conclusion requires more solid evidence.

In general, the tree construction needs a more careful handling of the sequence data. A closer look at some of the sequences revealed that the presence of cyclin and UDG motifs in the putative IF sequences is probably an artifact due to errors in the automatic gene predictions; e. g.: XP 003386754_1 and GAA48725_1 predict a lamin-cyclin chimera and a lamin-uracil−DNA glycosylase chimera, respectively. The existence of such chimeric proteins is not supported by expression data. Other sequences are incomplete precluding an assignment as either lamin or cytoplasmic IF. The sequence data used for the tree building should be carefully checked to avoid unnecessary confusion.

Additional data files and statistical comments:

1) The sequences of the proteins described in this study should be given in the supplementary information together with the accession numbers of the corresponding genomic contigs (at least for the three tardigrade proteins).

2) A sequence alignment with a selection of other lamin/cytoplasmic IF proteins would be desirable.

Reviewer #4:

This is an interesting paper which is based on two pieces of evidence.

First, the authors perform a detailed phylogenetic analysis of IF proteins and their relatives, including isomin. They conclude that isomin clusters with lamins rather than with cytoplasmic IFs.

Second, the authors experimentally explore the localization of three lamins (1,2 and 3) present in a tardigrade. They show that while 1 and 2 are located in the nucleus as expected, lamin-3 is observed in the cytoplasm of epithelial cells.

The main conclusion of the paper is that the true cytoplasmic IFs were lost in panarthropods, but lamin-like IFs can take over their functions in the cytoplasm. While this theory certainly needs additional proof (beyond the reported data on tardigrade lamin-3 and in fact on isomin as well), I think that this is an important advance.

The new findings also suggest that the current terminology may need some adjustment, as the name 'cytoplasmic IFs' becomes ambiguous as one reads through the paper. Should one speak of 'non-lamin' IFs vs. lamins to reflect the phylogeny, which would then be uncoupled from the cytoplasmic vs. nuclear localization?

The phylogenetic analysis of Mencarelli et al. was less rigorous than the one presented here. As far as I can tell from that paper, Mencarelli et al. simply claimed that isomin was an IF relative, found in the cytoplasm (just like the tardigrade lam3 discussed here). The new phylogenetic analysis is much more systematic, relating isomin to lamins.

Thus there does not seem to be a real contradiction between Mencarelli et al. and the current paper (as suggested in the first paragraph of the Results and Discussion section). I think that the authors should adjust this part of the text accordingly.

In any case, the new paper is an important step forward.

1) While it is clear that the full phylogenetic tree can only fit in the supplement, the authors should present some simplified but yet scientifically rigorous view of such a tree. The current drawing shown in Figure 1 is oversimplified. (In fact, the tree shown in Mencarelli et al. would be a good example).

2) Regarding Figure 1—figure supplement, it has been known for decades how coiled coils fold. Trying to model a structure of a monomer without taking into account the interaction between the chains (usually two) of the coiled coil clearly yields a nonsense result here. The authors should either re-do the modelling properly or remove the figure (which I believe is not important here).

---

## [Author Response]

Required revisions: The reviewers raise a number of concerns that should be appropriately considered before the paper can be accepted. Those comments concerning the sequences used to construct the "tree" may cause some extra work, but the remainder of the comments concern the improvement of the text, in particular the background of the research.

*1) The amino acid sequence of the α-helical rod provides a good starting point to characterize potential intermediate filament (IF) proteins by comparing them to authentic vertebrate ones. As the conserved end domains are evolutionary signatures with functional roles, those segments should be looked at in particular. Hence, an amino acid comparison of lamin-1 to lamin-2 to lamin-3 with each other and with e.g. human, C. elegans and Hydra or a protist lamin would be a meaningful way to analyze these new proteins before drawing domain schemes.*

We compared the tardigrade IF proteins with well-known and authentic IF proteins (lamins and cytoplasmic IFs) of *H. sapiens* and *C. elegans* and included these alignments (Figure 2 and Figure 2—figure supplement 1 and Figure 2—figure supplement 2) to better characterize the domain structure of the new tardigrade IF proteins.

2) With respect to the last comment, the work by Martin Kollmar (Scientific Reports 5:10652) has to be discussed appropriately.

We included the work by Kollmar (2015) in our Discussion and pointed out that irrespective of the proposed polyphyly of metazoan lamins, our conclusion of convergent evolution of cytoplasmic IFs in *H. dujardini* (cytotardin) and in collembolans (isomin) within the bilaterian lamins remains unaffected due to their nested (and solidly supported) placement in the tree.

3) The fact that cytoplasmic IF proteins (which are essentially not lamins) are derived from lamins is not so new as one would learn from the work over the last two decades of the Weber group in Göttingen. How can one leave out this worm in light of its close relatedness to water bears?

We enhanced our Discussion to better highlight the relatedness of the tardigrade IFs to IF proteins from *C. elegans*. In particular, although the domain composition and structure of the cytotardin resembles those of known cytoplasmic IFs from *C. elegans*, we show that, from an evolutionary point of view, the cytotardin from *H. dujardini* is more closely related to its lamins (or all other known bilaterian lamins) than to the known nematode cytoplasmic IFs. Hence, we suggest convergent evolution and neofunctionalization of cytoplasmic IFs at least in *H. dujardini* (but probably also in certain copepods and collembolans).

*4) A figure with an evolutionary "tree" that can be printed on one page would be good* – *as it would only allow giving main relationships. The Weber group did such overviews in their papers.*

We removed the oversimplified drawing in Figure 1. Instead we included a completely new and more descriptive figure of the tree, which also fits on a single page in the main text (Figure 3).

*5) The issue of cytoplasmic IFs* – *as well as isomin* – *may be discussed with a broader perspective. In the light of the million insect species, Drosophila is surely not "the" insect.*

We fully agree with the reviewers that *Drosophila* is not “the” insect. Hence, we aimed right from the beginning of the study to screen for intermediate filament proteins in as many distantly related arthropod taxa as possible. Besides the putative chelicerate, crustacean, and hexapod IF proteins obtained from publicly available databases (such as GenBank) we screened over 70 additional hexapod transcriptomes sequenced as part of the 1KITE project (Misof et al. 2014, Science 346, 763–767). To emphasize this, we revised the text of our manuscript accordingly (Results and Discussion, second paragraph).

*6) The sequences used to build the tree should be "cleaned".*

We removed the questionable sequences mentioned by Reviewer #3 and repeated all analyses with the cleaned dataset.

*7) IF proteins form* – *with very rare exceptions (Keratin 18)* –

*parallel coiled-coil dimers. Meaningful molecular modeling should consider that fact.*Following the reviewers’ advice we decided to remove the modeling. Instead, we describe the new tardigrade IFs by protein sequence comparison with authentic IF proteins from humans and nematodes (see also our response to the comment #1 above).

*Full Reviews Reviewer #1: Lars Hering and colleagues report on the occurrence of both nuclear and cytoplasmic intermediate filament (IF) proteins in Tardigrades, the water bear. Based on in silico work done for Hypsibius dujardini they derive two amino acid sequences that exhibit similarity to lamins and one that is cytoplasmic. As tardigrades are very special organisms, tolerating various extreme stresses, they are of high interest for every life scientist. These data they use to investigate evolutionary relationships. The evolutionary trees provided are interesting, but the definition of the proteins is poor (see below). But immuno-cytochemical methods the authors demonstrate that the two nuclear IF proteins localize to the nuclear envelope. Moreover, these authors use cDNAs to transfect human epithelial cells and demonstrate that the cytoplasmic lamin-3 (what a misnomer) assembles into some type of regular network. All the cell biological data are very interesting and of high quality.*We agree with the reviewer that the term “lamin” is mostly understood as a functional term rather than a taxonomic one from an evolutionary point of view. We therefore renamed lamin-3 into “cytotardin”, referring to the cytoplasmic nature of this IF.

For a reader following the DNA-RNA-protein dogma for the last decades, a protein is still defined by its sequence, and not its position in a phylogenetic tree. Old fashioned as it may be, it would be good to discuss such issues as domain size, coiled-coil forming ability, i.e. the occurrence of heptad or hendcad patterns. The sequences are, however, not shown neither are they directly compared with authentic lamins such as those from Hydra, man or C. elegans. Strange enough, the latter organism is not centrally discussed, as for this worm Klaus Weber and his group in Göttingen demonstrated how lamins and invertebrate cytoplasmic intermediate filament proteins are related, i.e. by loss of the nuclear localization signal (NLS) the cytoplasmic IF proteins emerged (Dodemont et al. 1990 & 1994 EMBO J., Andreas Erber et al. 1998 & J. Mol. Evol., and several more papers). Obviously a fruitful strategy. A selected paradigmatic-type comparison would be very instructive. I took the CAAX-box-free version and compared its sequence with human lamin A: 1. Coil 1B is missing 4 amino acids and coil 2 carries a 3-amino acid insertion. The size of these helical segments including coil 1A is highly conserved, with the rare exception of coil 2 of the single lamin of C. elegans, which is missing 2 heptads or 14 amino acids. Notably, the IF consensus motif in coil 1A exhibits high conservation as does the IF consensus motif at the end of the rod in coil 2. More surprisingly, the linker segment L1 is entirely kept both to the very amino acid number and with respect to the quality of amino acids, which all are good for α-helix formation; the same holds true for linker L12 from what one can conclude that structural principals in the α-helical rod domain are highly conserved. Figure 1 does not give any clue to how related these domains are. In Figure 1 the subdivision in 2A and 2B is completely illusory, it is wrong. The authors should refer to a recent structural review of Sergei Strelkov (Leuven) in Current Opinion in Cell Biology (2015).

Regarding the subdivision of the rod domain we are now following the recommendation of the mentioned review by the Strelkov lab and modified our manuscript accordingly.

"Head" and "tail" sequences vary very much in IF proteins – notably here: the Ig fold sequence in the carboxyl-terminal domain exhibits highest identity with that of human lamin A, when the whole amino acid sequence is considered. The NLS shows an interesting variation from the standard, both CDK1 sequences flanking the rod are there but slightly shifted compared with the position of lamin A. In summary, Figure 1 gives the impression that everything is like in every IF protein and that lamin-1 to -3 are identical: to what extent are they? The tree in Figure 1 does not tell anything in detail. So how different is the tardigrade cytoplasmic IF protein from its lamins, and how closely is it related to human keratins K8 and K18? Does it have features that are anything like "keratin-related"? Along these lines: the structural models shown as Figure 1—figure supplement 3 are pretty meaningless –

*all three lamin-1 to -3 should form coiled coils.*Since our main goal was not to clarify all structural details of the tardigrade intermediate filament proteins, we focused on the characterization based on the presence or absence of typical lamin features, including NLS and CaaX, in conjunction with the localization of these proteins within the tissues. In addition, we wanted to shed light on the evolutionary history of IF proteins within panarthropods, with special emphasis on the fate of cIFs in arthropods. Nevertheless, we now included alignments with well-known IF proteins (Figure 2 and Figure 2—figure supplement 1 and Figure 2—figure supplement 2) to better characterize the domain structure of the described proteins.

Moreover, we enhanced our Discussion to better highlight the relatedness of the tardigrade IFs to IF proteins from *C. elegans*. In particular, although the domain composition and structure of the cytotardin resembles those of known cytoplasmic IFs from *C. elegans*, we tried to show that, from an evolutionary point of view, the cytotardin from *H. dujardini* is more closely related to its lamins (or all other known bilaterian lamins) than to the known nematode cytoplasmic IFs. Hence, we suggest convergent evolution and neofunctionalization of cytoplasmic IFs at least in *H. dujardini* (but probably also in certain copepods and collembolans).

We also agree with the reviewer that Figure 1—figure supplement 3 is unnecessary and therefore we removed this figure. In addition, we removed the oversimplified tree in Figure 1. Instead, we now provide a new and more descriptive figure of the tree, which also fits on a single page in the main manuscript (new Figure 3).

*Additional data files and statistical comments: The amino acid sequences and the kind how coiled-coil forming regions are present and can/do define a domain should definitively be provided.*We included alignments with well-known IF proteins (Figure 2 and Figure 2—figure supplement 1 and Figure 2—figure supplement 2) to better characterize the domain structure of the described proteins.

Reviewer #3: Hering et al. report on the evolution of cytoplasmic intermediate filament proteins (cIF) in the panarthropod lineage. They have characterized three tardigrade IF proteins in more detail: two lamins and one cytoplasmic IF protein. They convincingly show by immunofluorescence microscopy that the two lamins are nuclear and that the third protein is cytoplasmic. It forms filamentous structures in the cytoplasm of epithelial cells. This part of the manuscript is clear and convincing. From their data they conclude that the genes for cytoplasmic IF proteins have been lost at the base of the panarthropod lineage and that cIF genes have been acquired at least three times independently in collembolans, copepods, and tardigrades. This is a very interesting point. This statement is based solely on evolutionary tree construction of a broad range of IF rod sequences. However, it is not fully convincing. For the collembolan lineage it is based solely on the fact that the isomin rod sequence groups with bilaterian lamin rod sequences. At none of the nodes of the two trees bootstrap values exceed 50%. Given that collembolans are at the base of the arthropod lineage this conclusion requires more solid evidence.

*In general, the tree construction needs a more careful handling of the sequence data. A closer look at some of the sequences revealed that the presence of cyclin and UDG motifs in the putative IF sequences is probably an artifact due to errors in the automatic gene predictions; e. g.: XP 003386754_1 and GAA48725_1 predict a lamin-cyclin chimera and a lamin-uracil−DNA glycosylase chimera, respectively. The existence of such chimeric proteins is not supported by expression data. Other sequences are incomplete precluding an assignment as either lamin or cytoplasmic IF. The sequence data used for the tree building should be carefully checked to avoid unnecessary confusion.*We excluded chimeric and incomplete sequences from our datasets and performed completely new analyses with these cleaned datasets.

*Additional data files and statistical comments: 1) The sequences of the proteins described in this study should be given in the supplementary information together with the accession numbers of the corresponding genomic contigs (at least for the three tardigrade proteins).*

All sequences obtained for this work were submitted to GenBank and the corresponding accession numbers are provided in the Material and methods section as well as in the supporting trees of our revised manuscript (Figure 3—figure supplement 1 and Figure 3—figure supplement 2).

*2) A sequence alignment with a selection of other lamin/cytoplasmic IF proteins would be desirable.*We included alignments with well-known IF proteins (Figure 2 and Figure 2—figure supplement 1 and Figure 2—figure supplement 2) to better characterize the domain structure of the described proteins.

Reviewer #4: This is an interesting paper which is based on two pieces of evidence. First, the authors perform a detailed phylogenetic analysis of IF proteins and their relatives, including isomin. They conclude that isomin clusters with lamins rather than with cytoplasmic IFs. Second, the authors experimentally explore the localization of three lamins (1,2 and 3) present in a tardigrade. They show that while 1 and 2 are located in the nucleus as expected, lamin-3 is observed in the cytoplasm of epithelial cells. The main conclusion of the paper is that the true cytoplasmic IFs were lost in panarthropods, but lamin-like IFs can take over their functions in the cytoplasm. While this theory certainly needs additional proof (beyond the reported data on tardigrade lamin-3 and in fact on isomin as well), I think that this is an important advance. The new findings also suggest that the current terminology may need some adjustment, as the name 'cytoplasmic IFs' becomes ambiguous as one reads through the paper. Should one speak of 'non-lamin' IFs vs. lamins to reflect the phylogeny, which would then be uncoupled from the cytoplasmic vs. nuclear localization?

Since the terms “lamin” and “cytoplasmic IF” are mostly understood as functional terms rather than taxonomic ones from an evolutionary point of view and to prevent confusion for the readers, we renamed lamin-3 into “cytotardin”, referring to the cytoplasmic nature of this IF.

We agree with the reviewer that the terminology is indeed ambiguous in general, depending on the readers’ personal point of view. From a functional perspective, all lamins occur in the nuclear lamina and all cytoplasmic IF within the cytoplasm, irrespective of their evolutionary history. But as we could show in the manuscript, some “functional cytoplasmic IFs” are more closely related to lamins than to other cytoplasmic IFs.

*The phylogenetic analysis of Mencarelli et al. was less rigorous than the one presented here. As far as I can tell from that paper, Mencarelli et al. simply claimed that isomin was an IF relative, found in the cytoplasm (just like the tardigrade lam3 discussed here). The new phylogenetic analysis is much more systematic, relating isomin to lamins. Thus there does not seem to be a real contradiction between Mencarelli et al. and the current paper (as suggested in the first paragraph of the Results and Discussion section). I think that the authors should adjust this part of the text accordingly.*Mencarelli et al. suggested that isomin is closely related to the cIFs from nematodes (which are indeed members of the monophyletic bilaterian cIF clade), which is – according to our analyses – not the case. Consequently, these authors concluded that “the expression of an IF terminal web in the midgut of *Isotomurus* species appears to be evolutionarily related to the original intestine organization present in the ancestor common to arthropods and nematodes”. This indeed contradicts our findings, since not a single IF homolog of panarthropods clusters within the bilaterian cIF clade (as the nematode cIFs and priapulid cIFs do). Nevertheless, we slightly rephrased our sentence to mitigate the tone.

*In any case, the new paper is an important step forward.1) While it is clear that the full phylogenetic tree can only fit in the supplement, the authors should present some simplified but yet scientifically rigorous view of such a tree. The current drawing shown in Figure 1 is oversimplified. (In fact, the tree shown in Mencarelli et al. would be a good example).*We removed the oversimplified drawing in Figure 1. Instead we included a completely new and more descriptive figure of the tree, which would fit on a single page of the main manuscript (new Figure 3).

2) Regarding Figure 1—figure supplement 3, it has been known for decades how coiled coils fold. Trying to model a structure of a monomer without taking into account the interaction between the chains (usually two) of the coiled coil clearly yields a nonsense result here. The authors should either re-do the modelling properly or remove the figure (which I believe is not important here).

We agree with the reviewer that this figure is unnecessary here and decided to remove it. Instead, we included alignments with well-known IF proteins (Figure 2 and Figure 2—figure supplement 1 and Figure 2—figure supplement 2) to better characterize the domain structure of the described proteins.